# Comparative Genome-Wide Analysis of Two *Caryopteris x Clandonensis* Cultivars: Insights on the Biosynthesis of Volatile Terpenoids

**DOI:** 10.3390/plants12030632

**Published:** 2023-02-01

**Authors:** Manfred Ritz, Nadim Ahmad, Thomas Brueck, Norbert Mehlmer

**Affiliations:** Werner Siemens Chair of Synthetic Biotechnology, Department of Chemistry, Technical University of Munich (TUM), 85748 Garching, Germany

**Keywords:** reference genome, terpene synthases, *Caryopteris x clandonensis*, plant volatiles, long read sequencing, TPS subfamilies

## Abstract

*Caryopteris x Clandonensis*, also known as bluebeard, is an ornamental plant containing a large variety of terpenes and terpene-like compounds. Four different cultivars were subjected to a principal component analysis to elucidate variations in terpenoid-biosynthesis and consequently, two representative cultivars were sequenced on a genomic level. Functional annotation of genes as well as comparative genome analysis on long read datasets enabled the identification of cultivar-specific terpene synthase and cytochrome p450 enzyme sequences. This enables new insights, especially since terpenoids in research and industry are gaining increasing interest due to their importance in areas such as food preservation, fragrances, or as active ingredients in pharmaceutical formulations. According to BUSCO assessments, the presented genomes have an average size of 355 Mb and about 96.8% completeness. An average of 52,090 genes could be annotated as putative proteins, whereas about 42 were associated with terpene synthases and about 1340 with cytochrome p450 enzymes.

## 1. Introduction

Throughout the last decades, terpenes and terpenoids became more and more important in industrial applications. In the food industry terpenes are used, e.g., as flavoring compounds [1] or preservatives [2]. Due to its plant origin, the acceptance as a food additive is higher compared to chemical synthesis. In a pharmaceutical context the research and use of essential oils—with terpenes as their main components—range from anti-inflammatory [3], and immunomodulatory [4] to antiviral [5] and further indications [6,7,8,9,10,11]. The anti-cancer drug Taxol consists of a diterpenoid backbone [12] and is employed in different cancer treatments [13]. The success of this terpenoid surely is one of the reasons to further research terpenoids for pharmaceutical applications. Along with these applications, this class of molecules can be found throughout most organisms. Flowering plants show a vast diversity of terpenoids, which is a unique characteristic of the class *Angiospermae* [14]. In plants, they are used as a defense mechanism against biotic (e.g., herbivores or pests) and abiotic influences (e.g., radiation or climate stress) [15]. An example of a defense mechanism against biotic stress is the insect repellent activity of volatiles, such as p-menthane-3,8-diol from *Corymbia citriodora* [16,17]. This compound shows activity against the yellow fever mosquito *Aedes aegypti* [18]. *Caryopteris x clandonensis* essential oils also harbor a biological activity against these insects [19]. However, for these plants, the active agent is not yet identified. Additionally, terpenoids function as attractors for pollinators or as a possibility for energy storage [14]. 

The extensive diversity of natural terpenes derives from the conserved evolution of terpene synthases (TPS) and terpene-modifying enzymes, such as cytochrome p450 enzymes [20,21]. Terpenes are divided into different classes defined by their backbone. The basis is two building blocks, isopentenylpyrophosphate (IPP) and dimethylallyl diphosphate (DMAPP) which are synthesized in plants via the mevalonate pathway. IPP is the activated form of an isoprene unit consisting of five C-atoms (C5), also called hemiterpene. These are connected to larger units forming monoterpenes (C10), sesquiterpenes (C15), diterepenes (C20) and higher terpene structures [22]. Further steps of increasing terpenoid diversity involve the promiscuity of TPS as well as the subsequent modification by cytochrome p450 enzymes, which may encompass hydroxylation, carboxylation, acetylation or peroxide linkage. Examples include the biosynthesis of p-menthane-3,8-diol [17], gibberellin [23], taxol [24], and artemisinin [25], respectively. This results in a vast pool of natural compounds which account for a multitude of possible applications [14,26].

In general, plant TPS are divided into eight subfamilies which are grouped into classes I, II and III. This separation is based on functional assessment, sequence likelihood and architecture of genes. Class I is comprised of copalyl diphosphate synthases (TPS-c), *ent*-kaurene synthases (TPS-e), other diterpene synthases (TPS-f) and lycopod specific (TPS-h). TPS-d is only included in class II, which is specific for *Gymnosperms*. Lastly, class III consists of TPS-a, cyclic monoterpene synthases and hemi-TPS (TPS-b) and acyclic mono-TPS (TPS-g), which are *Angiosperm* specific [27].

With the advent of state-of-the-art bioinformatic technologies, deciphering the molecular mechanisms involved in the formation of terpenoids has become significantly easier [28]. Furthermore, the possibility to produce terpenes recombinantly by means of biotechnological production systems, rather than chemical synthesis, makes it an ecological and cost-effective technology for the increasing demand for terpenes in industrial applications, despite open challenges [29].

The combination of cutting-edge bioinformatics and next-generation sequencing technologies provided by Pacific Biosciences, Oxford Nanopore and Illumina allows for the rapid generation of draft genomes as well as the annotation of valid gene models. In this context, long-read sequencing technologies will be highlighted, as they exhibit no amplification biases. Consequently, these technologies provide a reliable basis for de novo whole-genome assemblies. Openly accessible bioinformatic tools enable cost-efficient assemblies, annotations and secondary downstream analyses for a broad range of scientists, and are publicly available via www.github.com (accessed on 11 December 2022) [30]. Two of these are the Quality Assessment Tool for *Genome* Assemblies (QUAST) [31] and Benchmarking Universal Single-Copy Orthologs (BUSCO). The latter is employed to assess the completeness of the obtained genome assemblies. Here, conserved and species-specific gene sequences are curated in databases and detected via a match-making algorithm to check for the gene set completeness of the evaluated taxonomic group [32]. An investigated genome is classified as complete if respective single-copy orthologs are present in the assembly.

In this work, we present the genomes of two *Caryopteris x clandonensis* cultivars (Dark Knight and Pink Perfection) from the *Lamiaceae* family in high quality employing long-read sequencing. These plants display a wide range of different metabolic pathways in regard to terpenoid biosynthesis, as also seen in other plants of the order *Lamiales,* e.g., in *Jasminum sambac* [33]. To elucidate variations between these multivariate datasets a principal component analysis (PCA) was conducted. Based on evident differences in volatile compound composition the two cultivars, Dark Knight and Pink Perfection were compared on a genomic level. This submission will be the 12th whole genome sequence within *Lamiaceae*, consisting of about 4788 further species, making it a source for gene sequences and further experimental basis in plant and natural product focused biosynthesis research.

## 2. Results and Discussion

### 2.1. PCA Analysis of Volatile Compounds

Differences between the volatile compounds of four cultivars were investigated using a GC-MS Headspace analysis. Ten main volatile components visible between the cultivars were selected, predominantly monoterpenoids and sesquiterpenoids, which are listed in Table 1. It has already been shown that there is a variety of monoterpene synthases that are able to catalyze ionization and isomerization starting from geranyl diphosphate [34]. Furthermore, the analysis of the cultivars revealed that a switch between pinene and limonene-derived compounds took place, which was sparsely synthesized in the other plants. In Table 1, these compounds are marked with an asterisk, one (*) represents limonene-related terpenoids, and two (**) represents pinene-related terpenoids. This especially is visible in the C4-C6 shift compared to the limonene backbone as seen in pinene (C4 to C6, see Appendix A). Similar substances could be identified as investigated previously for this plant species [19]. 

As the plants are cultivars from *Caryopteris x clandonensis* a common base profile (e.g., caryophyllene, perillyl alcohol, sabinene, farnesene or campholenal) of volatiles was expected, see Appendix A, as has been shown for other plants and their cultivars [35,36]. In this study, distinct differences between Dark Knight, Good as Gold, Hint of Gold, as well as Pink Perfection, can be shown. 

To further investigate the variations in the compound profile found during the analysis, a principal component analysis (PCA) was performed (Figure 1). Good as Gold and Hint of Gold express high morphological and metabolic similarity (see Appendix A). This is also evident in Figure 1, as both cultivars are located close to each other. On the other hand, Dark Knight and Pink Perfection showed the highest deviation in volatile compound composition. Moreover, the switch between C1 and C6 as mentioned above results in an intriguing product spectrum. These data underline the variations between the cultivars and demonstrate a need for further investigations into the molecular makeup of underlying TPS and cytochrome p450 enzymes, which are key for generating the molecular diversity of plant-based terpenoid structures in plants [20]. Therefore, due to their distinct differences revealed in the PCA, the two cultivars, Dark Knight and Pink Perfection, were sequenced to elucidate genomic differences and identify unique and yet unknown genes.

### 2.2. Genome Sequencing and Quality Assessment

In Table 2, the sequencing metrics of the respective Sequel IIe runs are depicted. Details regarding sequencing quality reports can be found in Appendix A. Total bases were nearly twofold higher in Dark Knight than in Pink Perfection, the same as obtained HiFi reads and yield. However, the HiFi read length, read quality and number of passes are comparable in both sequencing runs. Deviations in sequencing parameters are closely related to utilized libraries and input DNA quality. As the read quality is well above Q20 both runs were subjected to further analyses. 

In this study, both genomes of Dark Knight and Pink Perfection were assembled using the IPA assembler with a consecutive duplicate purging and phasing step. A QUAST analysis was conducted to assess assembly contiguity (see Table 3). 

The number of assembled contigs diverged in both candidates (see Table 3). However, respective L50 values were small (13 for Dark Knight and 14 for Pink Perfection) compared to obtained N50 (8.2 Mb and 7.1 Mb respectively), which assures gene integrity with only low or no fragmentation. The total contig length of complete genomes corresponds to their size, which is comparable (3.44 to 3.66 × 10^8^ bp), and the same as seen for GC content (31.5% and 31.77%). Furthermore, genome size was calculated using a k-mer-based analysis, with a k-mer size of 20. Results support the haploid genome size of ~355 Mb and estimated a diploid genome, see Appendix A. Based on the calculated genome size the coverage of Dark Knight and Pink Perfection resembles 74 and 38, respectively. 

To assess the genome completeness and reliability of both genome sequences, a Benchmarking Universal Single-Copy Orthologs (BUSCO) analysis was performed (see Figure 2). Both genomes were compared to the kingdom *Viridiplantae* and the clades *Embrophyta* and *Eudicotidae*, respectively. The selection of these lineages was based on the increasing grade of affiliation and the different accompanying BUSCO gene sets (in former order). For closer clades, more concise sequences are necessary in order to be identified as complete. In our case, even more affiliated clades show less deviation of completeness than expected in comparison to *Viridiplantae*. As the genomes were compared to different BUSCO datasets, the obtained results were depicted after normalization in Figure 2 to enable a concise comparison. Assessed genome completeness from the closest related clade (*Eudicotidae*) was 96.6% for Dark Knight and 96.8% for Pink Perfection, which were also compared to reference genomes of *Salvia splendens* (92.1%) [37] and *Sesamum indicum* (95.1%) [38]. The latter were only compared with the *Viridiplantae* database with BUSCO v2.0.1 and v3.0, whereas our data were analyzed by BUSCO v5.3.2. This may have caused the difference between 425 and 1440 BUSCO datasets, as frequent updates of the gene sets are necessary to improve BUSCO analysis [39]. The reference genomes were chosen due to the high prevalence in BLAST searches [40,41] using *Caryopteris x clandonensis* sequences. *S. splendens* appears to harbor mostly complete and duplicated BUSCOs, whereas *S. indicum* shows comparable results to the new genomes of Pink Perfection and Dark Knight with a majority of complete and single-copy BUSCOs. To interpret BUSCO results, it is necessary to understand duplicated BUSCOs and their nature, as these can be of biological or technical origin. In eukaryotic genomes, divergences in haplotypes often lead assemblers to form duplicates of high heterozygosity regions, resulting in contiguity issues and obstacles in further evaluation steps, such as gene annotation [42,43]. To circumvent these issues, tools such as “purge_dups” are utilized to remove duplicate regions (haplotigs) from the assembly to assure genome contiguity [42]. A consecutive polishing of obtained contigs and haplotigs using phasing results in increased genome quality. Of the newly assembled genomes only 0.24–0.69%/0.71–2.67% are fragmented or missing, respectively. The absence of some BUSCO genes may be due to a loss of true genes or these may be existing as true gene duplications [43]. 

### 2.3. Evaluation of Structural Differences between Genome Assemblies

To concisely compare genomes, the collinear gene order also known as synteny or syntheny blocks needs to be assessed [44]. It plays an important role in visualizing matches between organisms [45].

Investigating the synteny between cultivar genomes shows their close relation. Here, factors such as low contiguity and fragmentation have an effect on the analysis and lead to high error rates [46]. In our case, previously performed evaluations assured high contiguity and low fragmentation. Mauve was used to perform a multiple sequence alignment and applied to generate synteny blocks (Appendix A) [47]. Connections between these blocks reveal the high similarity within both genomes. This is typical for plant breeding, as specific traits are inherited from previous generations leading to inversions, duplications, or truncations in gene sets [48]. Furthermore, marker synteny can be used for phylogenetic analyses of cultivar evolution [49]. Thus, the plant samples seem to be closely related to the species *Caryopteris x clandonensis.*

### 2.4. Gene Models and Functional Annotation

Gene models were computed using the presented genome assembly and a long-read IsoSeq database as hints via AUGUSTUS [50,51,52,53]. As a training set *Solanum lycopersicum* was chosen due to its ancestral relation to *Lamiaceae*. For the cultivars, a total of 52,865 (Dark Knight), and 51,315 (Pink Perfection) genes were predicted and resemble putative proteins. The Cluster of Orthologous Groups (COG) and Gene Ontology (GO) terms were evaluated for all cultivars. It is to mention that only ~81% of the predicted genes were annotated using COG and GO databases. Out of these ~30% are poorly characterized (Figure 3E) and only a fraction (30%) of those can be assigned with GO terms. In regard to the complete genomes, nearly 20% of the proposed gene models remain without an assigned function. Figure 3 shows the COG counts for the following categories: (3B) cellular processes and signaling (3C) information storage and processing (3D) metabolism and (3E) poorly characterized. Figure 3A combines all the aforementioned categories. The obtained results emphasize a strong similarity in the compared cultivars. Further data in regard to the exact amount of COG per category can be found in Appendix A. This finding is a further indicator of the completeness of the presented genomes, as different cultivars have a similar set of genes, only varying in small nucleotide polymorphisms or other structural variants, which distinguish them [54,55]. 

A closer look into the different groups reveals characteristic functions in the cultivars. Most genes identified and functionally annotated are associated with replication, recombination and repair, which make up about 20.5% of total annotated genes Figure 3D) followed by signal transduction mechanism (~8%) (Figure 3A). Plants are exposed to endogenous and exogenous stresses such as chemicals or UV-radiation which can significantly alter DNA, thus there is high importance for repair mechanisms [56]. High redundancy of those ensures the safe replication of DNA with almost no errors [57]. 

In Figure 3C, proteins related to the COG category secondary metabolites biosynthesis, transport and catabolism, rank in second place within metabolism (2.8%). This category harbors TPS and cytochrome p450 enzymes. However, proteins associated with carbohydrate transport and metabolism are most abundant in this group as they are important for general metabolism and backbone synthesis. 

Compared to about 29,458 with COG functionally annotated genes, 11,118 unique GO terms were assigned to 14,280 different genes (27% of total gene models). COG terms are ancestrally conserved regions, GO terminology in contrast proposes functional annotation of each hypothetical gene. A gene-set enrichment analysis was conducted with GO terms as a source for gene sets [58]. The following figures show the GO term clustering regarding the three main categories in plants: biological process (Figure 4), molecular function (Figure 5) and cellular components (Figure 6). For all three an analysis was conducted based on GO terms identified in Pink Perfection. Detailed data for Dark Knight and Pink Perfection can be found in Appendix A. The GO analysis was visualized using REVIGO [59]. Respective cluster position within the semantic space is irrelevant, as similar semantic terms are located in vicinity of each other in the plot [58].

In Figure 4, GO terms related to biological processes are depicted with their respective prevalence (dot size). In addition, some clusters with similar functions were grouped by circles into the main function of these GO terms, as can be seen, e.g., with “translation” in the bottom right corner. Incorporated into this cluster are the terms: protein modification process, DNA metabolic process, nucleobase-containing compound metabolic process, and protein metabolic process. The cluster organelle organization includes cytoskeleton organization, cytoplasm organization, and mitochondrion organization. Clustered with transport: ion transport, protein transport. The last cluster response to stress contains the GO terms response to a biotic stimulus, response to an abiotic stimulus, response to an external stimulus, and response to an endogenous stimulus. GO terms without clustering but still strongly prevalent in the PCA are biological, metabolic and biosynthetic processes. 

For the GO analysis of the category molecular function, only one larger cluster was formed, which is nucleic acid binding. It incorporates the functions of DNA binding, RNA binding, and nucleotide binding. The two main components in this category are molecular function and catalytic activity. 

GO analysis in the category of cellular components yielded as the main results, intracellular anatomical structure and cellular components, as well as genes related to the cytoplasm. However, no semantic clustering was feasible based on the annotated GO terms. 

### 2.5. Identification of Terpenoid Biosynthesis Enzymes

InterProScan predicts distinct protein domains and classifies them into families [60,61]. The seed files PF01397, PF03936 and IPR036965 are associated with TPS activity. In the annotated protein database, these seeds were used as homology motifs. For Dark Knight 43 and Pink Perfection 41 TPS were identified. The seed file IPR001128 is related to cytochrome p450 enzymes. Here, we were able to identify Dark Knight and Pink Perfection 1316 and 1363 sequences. Compared to other plants these findings are comparable, both for TPS and cytochrome p450 enzymes [62,63,64,65]. 

To investigate the similarity and the affiliation into TPS subfamilies regarding identified TPS, a phylogenetic tree was constructed. Analysis was based on multiple sequence alignment by Clustal Omega using default parameters (see Figure 7). To differentiate between TPS families, 55 selected sequences of representative plant species were utilized as anchor sequences along with putative TPS from Dark Knight and Pink Perfection; the root was *Physcomitrella patens* (adapted from [66]). The multicolored clades belong to the different TPS subfamilies and are used as references, for a more detailed overview see the Appendix A. Concise numbers of TPS subfamily distribution in both cultivars are shown in Table 4. The most prominent subfamilies are TPS-a (green), TPS-b (black) and TPS-c (purple), which is in line with the distribution in *Eudicots*, *Angiosperms* and land plants [67]. The subfamilies TPS-d and TPS-h are not present in the investigated cultivars. These findings are supported by the literature, as TPS-d clusters are derived from *Gymnosperm* species [63,68] and TPS-h are specific for *Selaginella moellendorffii* [67].

## 3. Materials and Methods

### 3.1. Plant Material

Four cultivars of *Caryopteris x clandonensis* were acquired from a local nursery (Foerstner Pflanzen GmbH, Bietigheim-Bissingen, Germany) and grown to maturity in the open in a warm, moderate climate zone. After maturity, healthy leaves and blossoms were sampled and snap frozen in liquid nitrogen and stored at −80 °C until preparation for transcriptome and genome sequencing. Fresh mature leaves were used for GC-MS headspace analysis of volatile compounds. 

### 3.2. GC-MS Analysis of Volatile Compounds

Fresh mature leaves were weighed in GC headspace vials and analyzed using a Trace GC-MS Ultra system with DSQII (Thermo Scientific, Waltham, MA, USA). Vials were incubated for 30 min at 100 °C and a TriPlus autosampler was used to inject 500 µL of the sample in split mode onto a SGE BPX5 column (30 m, I.D 0.25 mm, film 0.25 µm); an injector temperature of 280 °C was used. The initial oven temperature was kept at 50 °C for 2.5 min. The temperature was increased with a ramp rate of 10 °C/min to 320 °C with a final hold for 5 min. Helium was used as a carrier gas with a flow rate of 1.2 mL/min and a split ratio of 8. The MS chromatograms and spectra were recorded at 70 eV (EI). Masses were detected between 50 *m*/*z* and 650 *m*/*z* in the positive mode [69]. Samples were measured in biological triplicates and the area average was used to compare peaks. Compounds were identified by spectral comparison with a NIST/EPA/NIH MS library version 2.0. To provide insight into the differentiation between plant samples a PCA was conducted.

### 3.3. High Molecular Weight DNA Extraction and Library Preparation

High molecular weight genomic DNA (HMW gDNA) suitable for long-read sequencing was achieved using a plant-optimized CTAB—PCI extraction method based on different protocols [70,71,72]; 1 g of frozen, unthawed plant leaves were ground using a CryoMill (Retsch, Haan, Germany; three cycles, 6 min precool at 5 Hz, disruption 2:30 min 25 Hz, cooling between cycles 0:30 min at 5 Hz). A CTAB extraction buffer (2% CTAB, 100 mM Tris pH 8.0, 20 mM EDTA, 1.4 M NaCl) was supplemented with 2% PVP prior to usage and solved at 60 °C. The unthawed fine powder was mixed with 5 ml buffer and incubated with 200 µL Proteinase K (Qiagen, Venlo, The Netherlands) for 30 min at 50 °C and occasionally inverted. At room temperature, 1 mg RNAse A (Thermo Scientific, Waltham, MA, USA) was added and incubated for 10 min. The mixture was washed twice, saving and reusing the aqueous upper phase, with one volume PCI (25:24:1) and three times with chloroform (10,000× *g*, 5 min, 10 °C). To pellet the HMW gDNA, 30% PEG was added to the aqueous phase (1:4), inverted, incubated for 30 min on ice and spun for 30 min at 12,000× *g*, 10 °C. The resulting shallow and colorless pellet was washed three times with 70% ethanol (5000× *g*, 5 min, 10 °C) and consequently, air dried at 40 °C and resuspended with 100 µL elution buffer (Qiagen, Venlo, The Netherlands). Quality and size of the gDNA were assessed using a Qubit dsDNA HS Kit (Thermo Scientific, Waltham, MA, USA), a Nanodrop photometer (Implen, Munich, Germany) and a Femto Pulse system (Agilent, Santa Clara, CA, USA), respectively. If variations in DNA concentration between Qubit and Nanodrop were > 50% an AMPure PB bead clean up or an electrophoretic clean up using a BluePippin system (Sage Science, Beverly, MA, USA) was performed; 5 µg HMW gDNA were sheared in a gTube (Covaris, Woburn, MA, USA; 1700× *g*) and used for whole genome library preparation using SMRTbell prep kit 3.0 (Pacific Biosciences, Menlo Park, CA, USA) according to the manufacturer’s recommendations. Size selection of the resulting library was performed using AMPure PB beads. Libraries were stored at −20 °C. Prior sequencing, primer and polymerase were bound using a Sequel II Binding Kit 3.2 (Pacific Biosciences, Menlo Park, CA, USA) according to the manufacturer’s recommendations.

### 3.4. Genome Sequencing and Assembly

Sequencing was performed on a Sequel IIe (Pacific Biosciences, Menlo Park, CA, USA) with two hours pre-extension, two hours adaptive loading (target p1 + p2 = 0.95) to an on-plate concentration of 85 pM, and 30 h movie time. The initial de novo genome assembly was performed using SMRT Link (v11.0.0+, Pacific Biosciences, Menlo Park, CA, USA) which uses Improved Phased Assembly (IPA) [73]. After polishing, the contigs were divided into primary and haplotype-associated contigs using purge_dups [74]. 

The assembled sequences can be found within the National Center for Biotechnology Information (NCBI). BioSample accession number: Dark Knight SAMN32308289, Pink Perfection SAMN32308290.

### 3.5. RNA Long Read IsoSeq 

To increase the quality of the genome assembly, long-read transcripts were sequenced to add more depth and accuracy to the proposed gene models. For RNA extraction, frozen, unthawed leaves were ground using a CryoMill and an RNeasy Plant Mini Kit (Qiagen, Venlo, Niederlande). A Turbo DNA free Kit (Invitrogen) was used to further clean the RNA. The high-quality RNA was used to perform an IsoSeq library prep using SMRTbell prep kit 3.0 and Sequel II Binding Kit 3.2. (Pacific Biosciences, Menlo Park, CA, USA). 

### 3.6. Bioinformatic and Statistical Analysis

Gene models were prepared through AUGUSTUS [50,51,52,53] using genomic data and long-read transcriptomic data as hints. Quality and completeness of the genome were estimated with QUAST (v5.2.0) [31] and BUSCO (v5.3.2) [39,43,75,76]. NCBI BLAST (v2.12.0+) [40,41] and InterProScan (v5.54-87) [60,61] were computed on a local computational unit. This analysis provided an annotation that was the basis for the determination of distinct protein families, in this case, terpene synthases and cytochrome p450 enzymes. EggNOG Mapper (v2.1.5) was used to determine COG and GO terms. Statistical analysis and figures were conducted using R (v4.2.1, revigo [59] and cateGOrizer [77]. Synteny analysis was performed using Mauve [47] (v2.4.0) and Geneious Prime (Geneious). For k-mer analysis jellyfish (v2.3.0) [78] was used (k-mer size: 20). GenomeScope [79,80] was used for the visualization of k-mer frequencies. The following analyses were conducted using galaxy project [81]: BUSCO, QUAST, EggNOG, Jellyfish, and GenomeScope. If not further specified default parameters were used for analysis.

### 3.7. Identification of TPS and Cytochrome p450 Enzymes

Genes associated with these protein classes were found using InterProScan and the domain seed files IPR036965 (TPS activity) and IPR01128 (cytochrome p450 enzymes). The phylogenetic tree was constructed using a global alignment with Blosum62. As a genetic distance model, Jukes–Cantor was chosen along with Neighbor-Joining as the Tree building method. The outlier was *Physcomitrella patens,* XP_024380398. Software used: Geneious Prime (Geneious).

## Figures and Tables

**Figure 1 plants-12-00632-f001:**
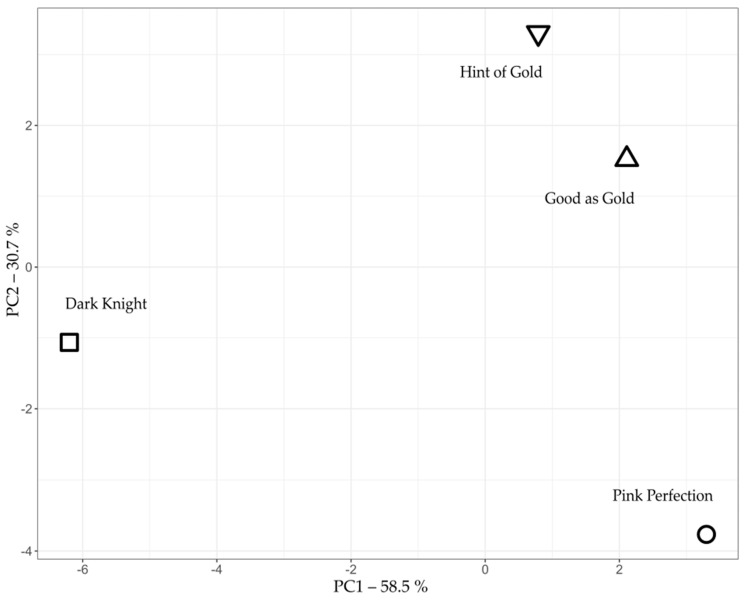
A principal component analysis of four different *Caryopteris x clandonensis* cultivars, Dark Knight, Good as Gold, Hint of Gold and Pink Perfection regarding the area of their volatile compounds analyzed by GC-MS Headspace.

**Figure 2 plants-12-00632-f002:**
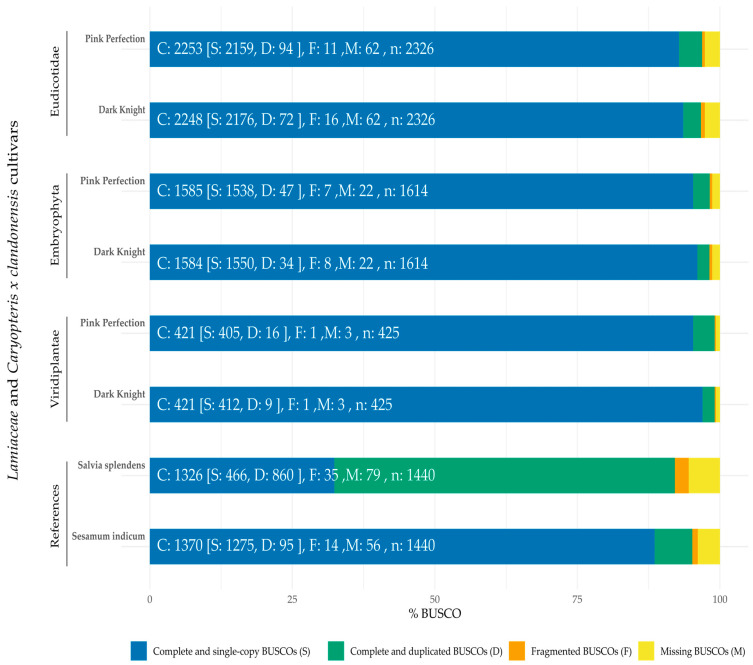
Comparison of BUSCO completeness of different cultivars of *Caryopteris x clandonensis* as well as *Salvia splendens* [37] and *Sesamum indicum* [38]. As the genomes were compared to other Benchmarking Universal Single-Copy Orthologs (BUSCO) datasets a normalization was performed to enable a comparison in genome completeness. Pink Perfection and Dark Knight were compared to the BUSCO datasets of *Viridiplantae*, *Embryophyta* and *Eudicotidae*, whereas *S. splendens* and *S. indicum* were compared to *Viridiplantae* only. Reference genomes were obtained from [37,38].

**Figure 3 plants-12-00632-f003:**
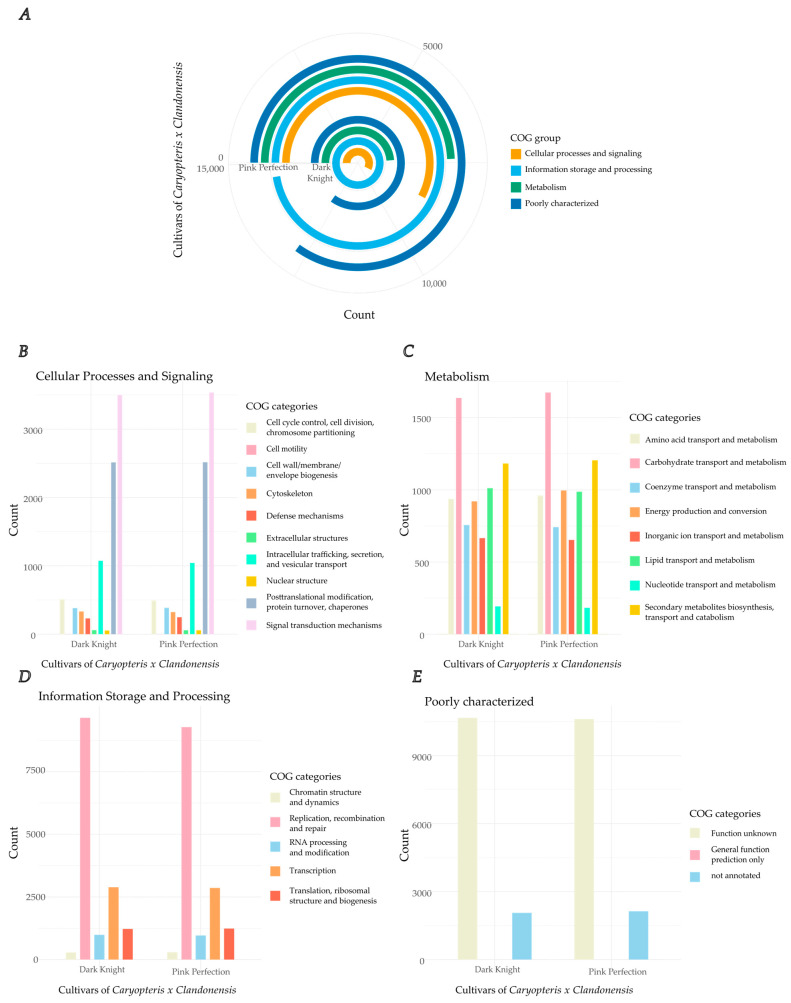
Annotation of gene sets for Cluster of Orthologous Groups (COG) for both cultivars, Dark Knight and Pink Perfection. (**A**) COG of two different cultivars of *Caryopteris x clandonensis*, Pink Perfection (outer ring) and Dark Knight (inner ring). Groups are divided in cellular processes and signaling, information storage and processing, metabolism, and a category for poorly characterized gene sets. (**B**) COG of cellular processes and signaling associated genes, total counts. (**C**) COG of metabolism-associated genes, total counts. (**D**) COG of information storage and processing associated genes, total counts. (**E**) COG of poorly characterized genes, total counts.

**Figure 4 plants-12-00632-f004:**
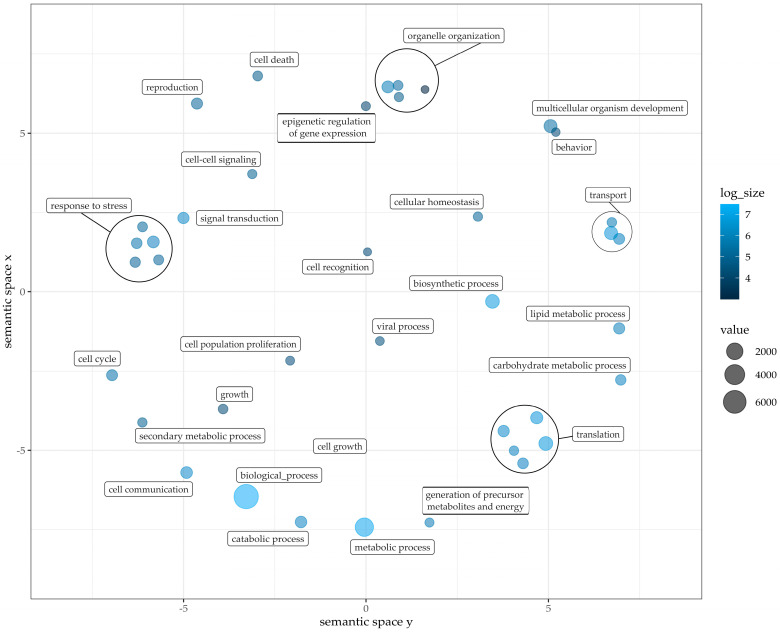
Gene Ontology term classification within biological processes of Pink Perfection. Clustered with response to stress: response to biotic stimulus, response to abiotic stimulus, response to external stimulus, response to endogenous stimulus. Clustered with translation: protein modification process, DNA metabolic process, nucleobase-containing compound metabolic process, protein metabolic process. Clustered with organelle organization: cytoskeleton organization, cytoplasm organization, mitochondrion organization. Clustered with transport: ion transport, protein transport. Figure was drafted employing REVIGO [59] and customized with R. Value and log size represents the counted GO terms across annotated gene models.

**Figure 5 plants-12-00632-f005:**
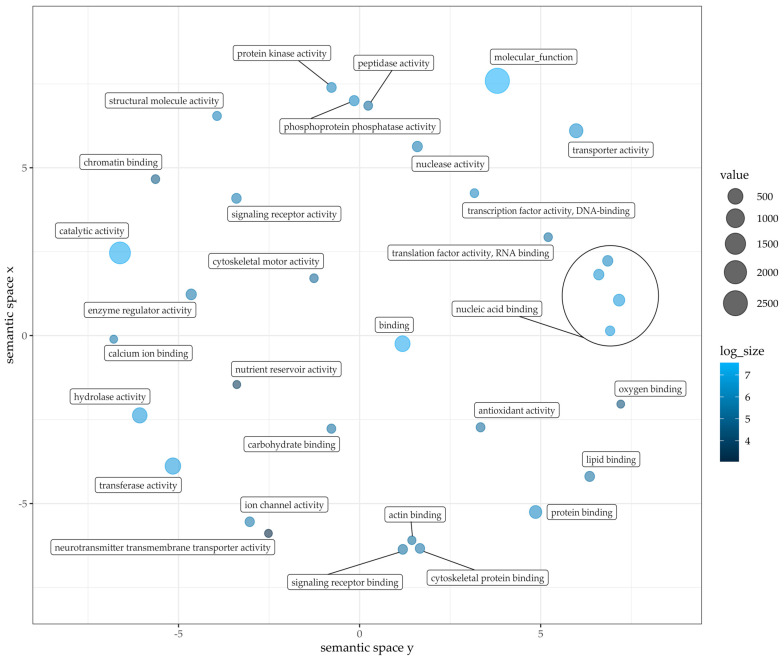
Gene Ontology term classification within molecular functions of Pink Perfection, clustered with nucleic acid binding: DNA binding, RNA binding, Nucleotide binding. Figure was drafted employing REVIGO [59] and customized with R. Value and log size represents the counted GO terms across annotated gene models.

**Figure 6 plants-12-00632-f006:**
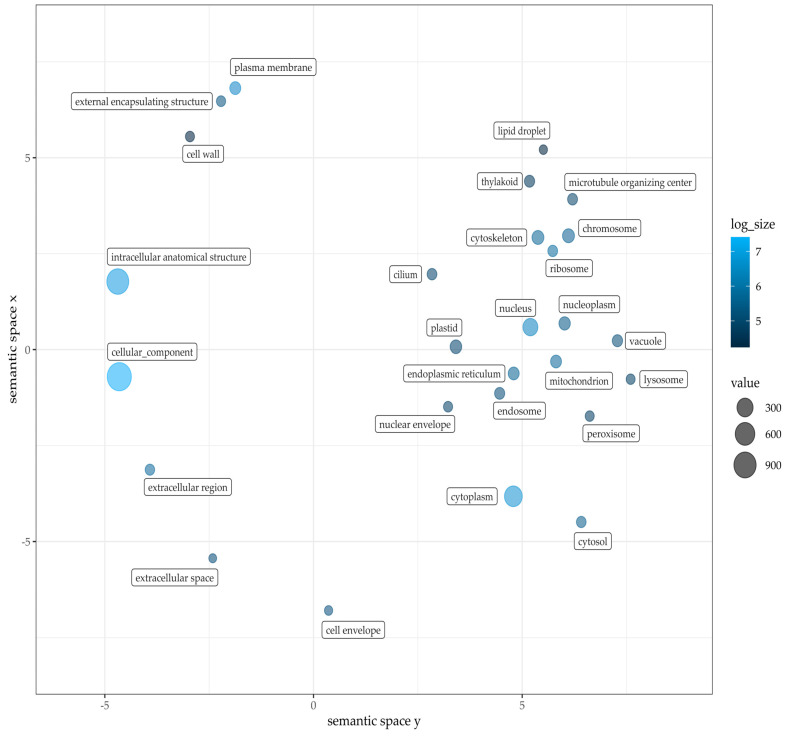
Gene Ontology term classification within cellular components of Pink Perfection. Figure was drafted employing REVIGO [59] and customized with R. Value and log size represent the counted GO terms across annotated gene models.

**Figure 7 plants-12-00632-f007:**
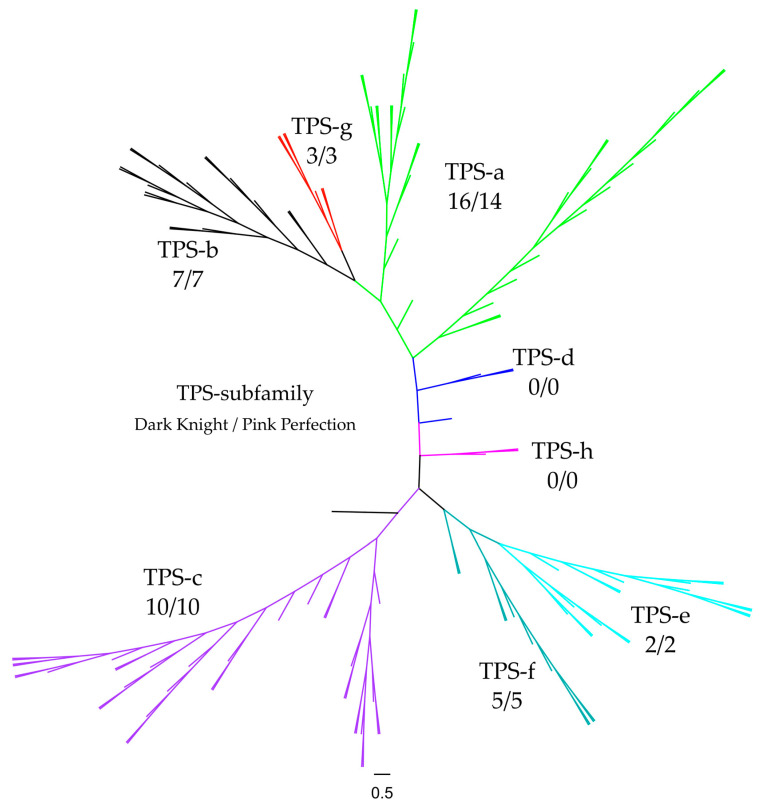
Phylogenetic tree of putative terpene synthases (TPS) within *Caryopteris x clandonensis* cultivars Dark Knight (DK) and Pink Perfection (PP). TPS-a (green), TPS-b (black), TPS-c (purple), TPS-d (blue), TPS-e (turquoise), TPS-f (petrol), TPS-g (red), TPS-h (pink). For phylogenetic tree construction, TPS a-h of selected plant species were included to assure correct classification of identified TPS. Numbers below the respective TPS subfamily indicate the count of predicted TPS in the genomes of the cultivars.

**Table 1 plants-12-00632-t001:** Ten main volatile compounds of four *Caryopteris x clandonensis* cultivars, visible between the cultivars were selected and are hierarchically listed (top: higher concentration, bottom: lower concentration). GC-MS Headspace was performed and an identification with a NIST/EPA/NIH MS library version 2.0 was conducted. * represents limonene-related terpenoids. ** represents pinene-related terpenoids.

Dark Knight	Good as Gold	Hint of Gold	Pink Perfection
α-pinene **	D-limonene *	D-limonene *	D-limonene *
trans-pinocarveol **	Cubebol	Cubebol	cis-p-mentha-1(7),8-dien-2-ol *
Pinocarvone **	Carvone *	trans-carveol *	trans-p-mentha-2,8-dien-1-ol *
Caryophyllene oxide	trans-carveol *	Carvone *	Caryophyllene oxide
β-pinene **	cis-p-mentha-1(7),8-dien-2-ol *	Caryophyllene oxide	trans-carveol *
(E,E)-α-farnesene	Caryophyllene oxide	trans-p-mentha-1(7),8-dien-2-ol *	cis-p-mentha-2,8-dien-1-ol *
α-campholenal	α-copaene	cis-p-mentha-1(7),8-dien-2-ol *	Carvone
α-copaene	β-pinene **	cis-p-mentha-2,8-dien-1-ol *	α-pinene **
Caryophyllene	cis-p-mentha-2,8-dien-1-ol *	α-copaene	β-pinene **
D-limonene *	trans-p-mentha-2,8-dien-1-ol *	trans-p-mentha-2,8-dien-1-ol *	Caryophyllene

**Table 2 plants-12-00632-t002:** Sequencing parameters of the PacBio Sequel IIe runs of *Caryopteris x clandonensis* cultivars Dark Knight and Pink Perfection.

Analysis Metric	Dark Knight	Pink Perfection
Total Bases (Gb)	444.13	229.43
HiFi Reads	1,823,939	843,632
HiFi Yield (Gb)	27.28	12.92
HiFi Read Length (mean, bp)	14,954	15,312
HiFi Read Quality (median)	Q35	Q34
HiFi Number of Passes (mean)	12	13

**Table 3 plants-12-00632-t003:** Genome contiguity assessment based on statistics generated by using QUAST.

Assembly	Dark Knight	Pink Perfection
# contigs	1183	782
Largest contig	29,672,976	31,977,049
Total length	366,625,098	344,117,456
Estimated reference length	300,000,000	300,000,000
GC (%)	31.50	31.77
N50	8,177,750	7,086,741
L50	13	14
# N’s per 100 kbp	0.41	0.44

**Table 4 plants-12-00632-t004:** Terpene synthase (TPS) subfamilies and their distribution in the *Caryopteris x clandonensis* cultivars Dark Knight and Pink Perfection. TPS-a, -b and -c show the highest prevalence in both cultivars.

TPS Subfamily	Dark Knight	Pink Perfection
a (green)	16	14
b (black)	7	7
c (purple)	10	10
d (blue)	-	-
e (turquoise)	2	2
f (petrol)	5	5
g (red)	3	3
h (pink)	-	-

## Data Availability

Data available in a publicly accessible repository. The data presented in this study are openly available in National Center for Biotechnology Information (NCBI). BioSample accession number: Dark Knight SAMN32308289, Pink Perfection SAMN32308290.

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
