# Peer review of "Comparative Genome-Wide Analysis of Two Caryopteris x Clandonensis Cultivars: Insights on the Biosynthesis of Volatile Terpenoids"

_plants, 2023, doi:10.3390/plants12030632_

Round 1

Reviewer 2 Report

This paper describes comparisons of terpenoid profiles in four cultivars of Caryopteris x clandonensis. It also reports on genome assemblies and gene annotations for two of the cultivars, and compares terpene synthase gene annotations for the two cultivars. The intended story here seems to be about comparative genomics of plant terpene synthases. However, the story is not clearly presented. There is plant phenotyping via volatiles analysis, and there is genome assembly/annotation/comparison, but the two are not very well connected. The flow of ideas is unfortunately haphazard. Some details of genome assembly and annotation are over-emphasized, and don’t add substance or explain the results. Figures are not presented in a logical order, and some are not legible. It is possible that major revision could yield a publishable product.  

Specific comments:

The introduction section is mostly about terpene biosynthesis, but doesn’t give much attention to the plants producing those molecules. The species that is the focus of this study isn’t mentioned until the last paragraph, and there it still isn’t clear whether this species has any economic importance as a crop, either for flavor/aroma compounds in consumer products, or as a medicine. Many genera with the Lamiaceae exhibit terpenoid diversity. What is notable about Caryopteris x clandonensis? Why were the four cultivars chosen for this study? Are any of them related by pedigree? The results section mentions “a common base profile (e.g. caryophyllene, perillyl alcohol, sabinene, farnesene or campholenal) of volatiles was expected…” The introduction section should have information about this, and mention any previous studies that have guided these expectations. A schematic of the most relevant branches of the biosynthetic pathway would be very helpful to tell the story, as most readers will not understand the relative positions of individual constituents within the pathway.

The results section is lacking in details. For example, Table 1 doesn’t show the relative quantities of the different volatiles. What percentage of the total volatiles in each plant does each constituent represent?  

Figures are not numbered according to the order in which they are cited. The first figure cited is supplemental figure S6, which really should be in the main body of the paper rather than in the supplemental section.

The description of the assembled genomes is inadequate. The haploid number of chromosomes in these plants is not mentioned, nor is the ploidy level. There is no estimate of genome size from any lab bench method such as flow cytometry, nor from any in silico analysis, such as a kmer-based analysis. The lack of such information makes it hard to assess the genome assembly.

The sequencing run metrics are shown in Table 2, but the genome assembly metrics are barely mentioned in the results text, and are only shown in Table 3.  The BUSCO results, Figure 2, are presented before the genome assembly metrics in Table 3, which seems backward.

The synteny analysis seems important, but it is buried in the supplemental section. Figure 3 is impossible to read with such small text.

The choice of tomato as a source for a gene annotation training set is strange, when there are genomes available for more closely related species within other Lamiaceae genera. The lack of GO terms for many genes is not surprising or unusual in plant genome annotations. However, the choice of a gene model training set from a genome within the Lamiaceae may improve gene annotation results overall. Figures 4 and 5 don’t add anything of substance to support the assertions about genome assembly quality, or to the terpenoid story.  

The manuscript needs careful proofreading. For example, ‘Pink Perfection’ is misspelled in Table 2.

Reviewer 3 Report

This manuscript reports the genome sequencing of two cultivars of Caryopteris x Clandonensis, focusing on the biosynthesis of volatile terpenoids. Overall, this is an important work and the sequenced genomes provide a new genetic source for volatile compound biosynthesis research in plants. I only have some minor comments as follows.

1. The authors selected “Dark Knight” and “Pink Perfection” as the two cultivars of Caryopteris x clandonensis for genome sequencing. Besides volatile compound composition, is there any other different traits between these two cultivars? Does “Dark” and “Pink” mean different colors of the two cultivars? The authors should provide the pictures of the two cultivars “Dark Knight” and “Pink Perfection”, as well as “Good as Gold” and “Hint of Gold”.

2. Based on differences in volatile compound composition of the two cultivars by GC-MS analysis, the authors focused on identification of terpenoid biosynthesis enzymes, trying to elucidate variations in TPS sequences of two cultivars compared on a genomic level. However, these TPS sequences in genomes are DNA sequences, not RNA sequences. In other words, different expressed RNA sequences make the difference of terpenoid biosynthesis and different products of volatile compounds, even though the DNA sequence is the same (due to epigenetic regulation). Thus, if the authors want to clarify the genomic differences between the two cultivars, the transcriptome sequencing is needed.

3. Since the manuscript is about genome sequencing and volatile terpenoids as well as TPS subfamilies in aromatic plants, I provide a recent published genome of Jasminum sambac for the author's reference.

Chen et al., The Jasmine (Jasminum sambac) Genome Provides Insight into the Biosynthesis of Flower Fragrances and Jasmonates. Genomics, Proteomics & Bioinformatics. doi: https://doi.org/10.1016/j.gpb.2022.12.005

Round 2

Reviewer 2 Report

This manuscript is much improved since the previous submission. Most of the previous comments have been adequately addressed. The figures are much more readable. There are a few details that still need correction prior to publication:

Line 81-82: “Here, species specific markers are curated in databases and detected via a match making algorithm.”

This statement is not entirely accurate. BUSCO databases do not contain genetic marker data; they contain gene sequences for sets of genes known to be conserved among different species within particular taxonomic groups.

Line 320 has an error related to a citation.

Author Response

Dear Editor, dear Reviewer,

Please find enclosed the revised manuscript entitled “Comparative genome-wide analysis of two Caryopteris x Clandonensis cultivars: Insights on the biosynthesis of volatile terpenoids” by Manfred Ritz and Nadim Ahmad et al. in your Special Issue “Applications of Bioinformatics in Plant Resources and Omics”.

We again, appreciate the time and effort you and each of the reviewers have dedicated to review the revised version of our first submission.

For the minor revision mentioned, we changed the wording in Line 81-82 as suggested by reviewer 2 to provide a clearer description of the basis of BUSCO databases.

Moreover, we checked the cross reference mentioned in line 320, which is now working correctly. We thank the reviewer for pointing this out and the thorough review including his feedback.

We hope the revised version is now appropriate for publishing in Plants, special issue “Applications of Bioinformatics in Plant Resources and Omics”.

Yours sincerely,

PD Dr. habil. Norbert Mehlmer